# Concentration of Polycyclic Aromatic Hydrocarbons (PAHs) in Human Serum and Adipose Tissues and Stimulatory Effect of Naphthalene in Adipogenesis in 3T3-L1 Cells

**DOI:** 10.3390/ijms24021455

**Published:** 2023-01-11

**Authors:** Ewa Mlyczyńska, Alice Bongrani, Christelle Rame, Małgorzata Węgiel, Anna Maślanka, Piotr Major, Piotr Zarzycki, Pierre-Henri Ducluzeau, Arnaud De Luca, Celine Bourbao-Tournois, Pascal Froment, Agnieszka Rak, Joëlle Dupont

**Affiliations:** 1Laboratory of Physiology and Toxicology of Reproduction, Institute of Zoology and Biomedical Research, Jagiellonian University in Krakow, Gronostajowa 9 Street, 30-387 Krakow, Poland; 2INRAE UMR 85 Physiologie de la Reproduction et des Comportements, 37380 Nouzilly, France; 3CNRS UMR 7247 Physiologie de la Reproduction et des Comportements, 37380 Nouzilly, France; 4Department of Animal Physiology, Université de Tours, 37041 Tours, France; 5Faculty of Chemical Engineering and Technology, Cracow University of Technology, Warszawska 24 Street, 31-155 Cracow, Poland; 62nd Department of General Surgery, Jagiellonian University Medical College, Macieja Jakubowskiego 2 Street, 30-688 Krakow, Poland; 7CHRU of Tours, Department of Endocrinology-Diabetology and Nutrition, 37032 Tours, France; 8Nutrition, Growth and Cancer (N2C) UMR 1069, University of Tours, INSERM, 37032 Tours, France; 9Department of Digestive Surgery, University Hospital of Tours, 37170 Tours, France

**Keywords:** polycyclic aromatic hydrocarbons, naphthalene, adipose tissue, 3T3-L1 cells, adipokines

## Abstract

Polycyclic aromatic hydrocarbons (PAHs) are one of the most prevalent classes of environmental pollutants. Some evidence shows that PAHs could be involved in human obesity. However, little is known about the distribution patterns of PAHs in human adipose tissue (AT) and the role of PAHs on adipogenesis/lipogenesis. The aims of this pilot study were to determine concentrations of 16 PAHs defined as high-priority pollutants in the plasma and adipose tissue of French and Polish bariatric patients, as well as their correlation with body mass index (BMI), plasma and AT adipokines expression levels. We finally investigated the role of naphthalene on cell proliferation, viability, and differentiation in 3T3-L1 preadipocytes. The concentration of most PAHs was similar in the three types of AT and it was significantly higher in AT as compared to plasma, suggesting bioaccumulation. Polish patients had higher PAH levels in AT than French ones. Only the concentration of naphthalene in AT was positively correlated with the BMI and serum or adipose chemerin, adiponectin and resistin expression, in French but not in Polish patients, who had significantly higher BMIs. Moreover, naphthalene exposure increased the cell proliferation of 3T3-L1 preadipocytes and lipogenesis, and increased the expression of genes involved in adipogenesis after cell differentiation. Taken together, PAHs and more particularly naphthalene could be an obesogenic molecule and increase the risk of obesity.

## 1. Introduction

Polycyclic aromatic hydrocarbons (PAHs) are widespread environmental pollutants with various adverse effects on human health. People are exposed to these compounds mainly through airborne emissions during incomplete combustion of organic materials, such as fossil fuels from industrial sources, automobile exhaust and, cigarette smoke, as well as through their intake with food [1,2,3]. Among hundreds of known organic compounds belonging to the PAHs group, the Environmental Protection Agency identified 16 PAHs as high-priority pollutants. This list includes the following: naphthalene, acenaphthene, phenanthrene, fluorene, anthracene, fluoranthene, pyrene, benzo(a)anthracene, chrysene, acenaphthylene, benzo(b)fluoranthene, benzo(k)fluoranthene, benzo(a)pyrene (B(a)P), indeno(1,2,3-cd)pyrene, dibenzo(a,h)anthracene, and benzo(g,h,i) perylene [4].

Despite the efforts to reduce PAH emissions in the air, these substances are still ubiquitous contaminants, especially in urban and industrial areas. Thus, exposure to PAHs is unavoidable [5]. Regular monitoring of air quality by governmental and non-profit organizations indicates differences in the degree of environmental pollution between countries resulting from economic and climate policies [6]. Therefore, citizens around the world are exposed to their bioaccumulation and action to varying degrees, which in the long-term perspective may result in different health effects in a given population. PAHs and their metabolites have been detected in many biological fluids including blood [7,8,9], urine [10,11,12,13], follicular fluid [7], sperm [14], and other biological samples [15].

Years of research confirm that PAHs have a negative impact on human health. They are carcinogenic and mutagenic factors [16], as well as known endocrine disruptors [17,18]. PAH exposure increases the risk of lung and cardiovascular disease [19,20,21] disrupts the reproductive process [22,23,24,25] and has many other negative effects [21]. Additionally, some evidence suggests a positive correlation between PAH concentration and obesity risk, indicating that PAHs may be obesogenic factors [11,13]. For example, epidemiological studies among children and adults show that urinary excretion of PAHs is positively associated with body mass index (BMI), waist circumference (WC), or waist-to-height ratio [13,26,27,28].

Obesity is currently considered as a pandemic of the 21st century, which affects millions of people around the world. Although a diet high in fat and sugar is the main factor contributing to obesity development [29], environmental factors including pollution are emerging other important contributors. Due to their lipophilicity, PAHs have the potential to be absorbed and accumulated by living organisms. So far, several studies have shown that PAHs can accumulate in rodents [30,31] pigs, cows [32] and human adipose tissue (AT) [33,34,35]. Moreover, it is known that some of them may change the levels of promoter methylation of genes involved in glucose and lipid metabolism, especially the insulin receptor gene, contributing to the pathogenesis of insulin resistance [36]. However, to date, there are no data about the direct effect of PAHs on the proliferation and differentiation of adipocytes as well as on their endocrine function. Indeed, AT is nowadays considered as a real endocrine organ secreting large amounts of biologically active molecules globally named adipokines. The most known are adiponectin, leptin, chemerin, visfatin and resistin [37]. These proteins regulate several physiological processes in AT, such as adipocyte proliferation and differentiation, insulin sensitivity, glucose uptake and lipid accumulation [38,39,40], thus representing biomarkers of AT function and potentially, of PAHs accumulation in this organ. 

Although the literature indicates that PAHs can be accumulated in AT and are associated with the risk of developing obesity, so far, the biological effects of these compounds on AT physiology are not fully understood. Therefore, the aim of the present study was firstly, to investigate the concentration of sixteen priority PAHs in plasma and in three types of AT, mesenteric, omental (epiploon) and subcutaneous, from French and Polish bariatric patients. Indeed, pollution levels in these two countries and particularly in the Malopolska (Poland) and Région Centre Val de Loire (France) areas are different, with higher levels of PAHs in Poland [41,42], which can affect the bioaccumulation of these compounds in human tissues. Next, we analyzed the correlation between the levels of the four most abundant PAHs with BMI and adipokine concentrations in plasma and AT. Finally, we determined the dose-dependent effects of naphthalene on the viability, proliferation and differentiation of murine 3T3-L1 preadipocytes, as well as its effects on their endocrine function.

## 2. Results

### 2.1. Characteristics of Bariatric Patients

For our study, we selected 10 bariatric male patients from France and 10 age-matched patients from Poland. Patient characteristics are detailed in Table 1. All patients presented with class III obesity, also referred to as severe, extreme, or massive obesity [43]. However Polish patients had significantly higher BMIs than French patients (*p* = 0.011, Table 1). Measurements of plasma glucose, insulin and the most important plasma adipokines like chemerin, adiponectin and visfatin showed no differences between the two groups. A significantly higher concentration was noted only for resistin in the plasma of Polish patients (*p* = 0.002, Table 1). French and Polish non diabetic patients displayed normal fasting glycemia (4.0 to 5.4 mmol/L) and plasma insulin levels (<174 pmol/L).

### 2.2. PAHs Concentrations in Blood Plasma and AT

The presence of all 16 priority PAHs was noted in plasma and three types of AT in both patient groups. The six most abundant compounds were naphthalene, phenanthrene, fluoranthene, pyrene, fluorene, and anthracene (Table 2 and Table 3). In general, the level of tested PAHs, except for pyrene in French patients, was lower in plasma compared to ATs (*p* < 0.0001, Table 2). Additionally, the concentration levels of anthracene, fluoranthene and benzo(a)pyrene in French patients and those of indeno(1,2,3-cd)pyrene in Polish group were similar in the three ATs and plasma. In Polish patients, for some compounds, benzo(b)fluoranthene, benzo(k)fluoranthene, and dibenz(a,h)anthracene, the concentrations did not differ significantly between plasma and subcutaneous AT as well between plasma and mesenteric AT for benzo(a)pyrene, benzo(g,h,i)perylene and dibenz(a,h)anthracene concentrations. Comparing the levels of PAHs in the three types of ATs, we observed that they did not differ significantly, except for in the case of phenanthrene in the group of French patients and pyrene in Polish patients; its level was markedly higher in subcutaneous compared to other types of AT (*p* < 0.0001, Table 2 and Table 3). 

### 2.3. Comparison of Plasma and AT PAHs Concentrations between French and Polish Patients

Comparing the PAH concentrations in plasma and ATs between two groups of patients, we observed significantly higher levels of the tested compounds in Polish patients, especially in the AT. In plasma, these differences were significant only for naphthalene, benzo(a)pyrene and dibenz(a,h)anthracene, with pyrene and benzo(a)pyrene being compounds with higher levels in French patients (*p* < 0.0001, Figure 1A,H,M,O). Regarding the PAH levels in the ATs, they were mostly and significantly more abundant in the tissues of Polish patients. We did not observe any differences except for naphthalene and benzo(b)fluoranthene in mesenteric AT, bezno(a)pyrene in mesenteric and omental AT and in all tissue in the case of indeno(1,2,3-cd)pyrene (Figure 1A,K,M,N).

### 2.4. Correlation between PAHs Concentration in ATs or Blood Plasma and BMI and Adipokines Level in French and Polish Patients

Based on the obtained concentration of PAHs, we selected the four most abundant compounds, naphthalene, phenanthrene, fluoranthene and pyrene, and checked whether their bioaccumulation in plasma and ATs correlated with the patients’ BMI and the level of several adipokines, which are indicators of adipose tissue function. We observed a statistically significant correlation with BMI only for the concentration of naphthalene in mesenteric (r = 0.743, *p* = 0.014, Table 4), omental (r = 0.904, *p* = 0.0003, Table 4) and subcutaneous ATs (r = 0.906 *p* = 0.0003, Table 4) in patients from France. Similarly, in these groups of patients, the concentration of naphthalene in omental AT strongly positively correlates with the concentration of plasma chemerin (r = 0.979, *p* < 0.0001, Table 5) and resistin (r = 0.838, *p* = 0.002, Table 5) while correlating negatively with adiponectin (r = −0.938, *p* < 0.0001, Table 5). These results were consistent with those obtained in omental AT, where we observed a significant correlation between naphthalene levels and RARRES2 (r = 0.679, *p* = 0.031, Table 6), RETN (r = 0.667 *p* = 0.035, Table 6), as well as ADIPOQ mRNA expression levels (r = −0.679, *p* = 0.031, Table 6). Interestingly, any correlation found was observed in Polish patients (Table 5 and Table 6).

### 2.5. Dose-Dependent Effects of Naphthalene on 3T3-L1 Cell Viability and Proliferation

Since naphthalene was the most abundant compound in French patients’ adipose tissues and positively correlated with BMI and plasma adipokine levels, we chose this compound to further investigate the effect of PAHs on 3T3-L1 pre-adipocyte functions. All doses of naphthalene (1, 10, 100 and 500 ng/mL) after 24 h of exposure significantly increased the viability of the 3T3-L1 cells, as tested by the CCK8 assay (*p* < 0.05, Figure 2A). Additionally, cell proliferation was also markedly higher after treatment with 10, 100 and 500 ng/mL as measured by the incorporation of BrdU (*p* < 0.05, Figure 2B).

### 2.6. Effects of Naphthalene on 3T3-L1 Cell Differentiation

Adipogenesis is the process of differentiation of pre-adipocytes into mature adipocytes, during which lipid droplets are accumulated [44]. In our study, staining of lipid droplets with Oil Red O powders in differentiated 3T3-L1 cells showed that treatment with 500 ng/mL of naphthalene increased the number of lipid droplets almost three-fold (*p* < 0.00001, Figure 3A,B). A dose of 10 ng/mL also had the tendency to increase the lipid content, but it was not statistically significant (*p* = 0.089, Figure 3A,B). Since adipocyte differentiation is a complex process controlled by many genes, we also examined naphthalene’s effect on mRNA expression of various proadipogenic and lipogenic factors. We noted that naphthalene at doses of 10 and 500 ng/mL markedly stimulated in a dose-dependent manner the expression of all tested genes: *Fasn, Acaca, Cebpa, Pparg, Fabp4, Plin1, Rarres2, Adipoq,* and *Retn* (*p* < 0.0001, Table 7). Additionally, we observed that both 10 and 500 ng/mL naphthalene upregulated tge secretion of chemerin by 3T3-L1 cells into the culture medium compared to controls (*p* < 0.05, Figure 4).

## 3. Discussion

In the current study, we detected the presence of all 16 priority PAHs in the blood plasma and AT of bariatric patients from France and Poland. We showed higher concentrations of PAHs in ATs than in blood plasma suggesting an adipose accumulation. Moreover, the concentration of most of the PAHs was similar in mesenteric, omental and subcutaneous AT but more important in Polish than in French patients. Interestingly, in French patients, we detected a positive correlation between naphthalene concentration in three ATs and BMI. Furthermore, naphthalene concentration in omental AT was positively correlated with chemerin plasma concentration and omental RARRES2 gene expression, suggesting a potential biomarker of chemerin of naphthalene exposure in AT. We also showed that a naphthalene exposure in 3T3-L1 preadipocytes increased cell proliferation and it improved lipid content and raised the gene expression of the main adipogenesis and lipogenesis markers during cell differentiation. Moreover, it increased RARRES2 mRNA expression and chemerin secretion in the culture medium, suggesting that naphthalene could accelerate the preadipocyte maturation and stimulate the expression of mature adipocyte proteins. 

### 3.1. Accumulation of PAHs in Human Adipose Tissues and Difference between French and Polish Patients

PAHs have a high ability to accumulate in human organs. So far, their presence has been demonstrated in the brain, liver, kidneys, lungs, heart, spleen, placenta and fat [15,23] as well as in biological fluids such as blood [7,8,9], follicular fluid [7], sperm [14] and urine [10,11,12,13]. Most studies, however, indicate that due to their lipophilic nature, PAHs are particularly accumulated in AT [15,34]. In our study, we also observed that the concentration of most of the tested compounds was higher in AT than in blood plasma. By accumulating PAHs, AT may have a protective effect on other organs, to limit excessive accumulation of these compounds in other tissues [45]. So far, there is evidence that many persistent organic pollutants, such as dioxins and polychlorinated biphenyls, accumulate in adipocytes, causing an increase in body fat content [46,47]. However, a problem may occur during weight loss, when the compounds accumulated in AT will be released into the bloodstream during fat burning or after bariatric surgery. Then, AT will be a continuous internal source of PAHs [45]. This is particularly dangerous considering that PAHs disrupt metabolic processes and hormonal balance, cause inflammation and, in the long term, are carcinogenic [16,17,21]. Considering that AT is not a homogeneous structure, we examined fat samples from different depots: mesenteric, omental, and subcutaneous. As it turns out, the location of AT determines its function; subcutaneous adipocytes have increased lipid metabolism activity and vesicular transport and secretion, which would also suggest that accumulated substances such as PAHs may be released to a greater extent to blood. On the other hand, visceral adipocytes have a higher expression of proteins involved in translational or biosynthetic activity and energy metabolism [48]. Interestingly, in our study, PAH concentrations in the three types of AT did not differ significantly, although only in the case of phenanthrene in French patients and pyrene in Polish patients were the levels higher in subcutaneous AT. Thus, PAHs accumulated in different ATs to a similar extent. We observed that the concentration of the main detected PAHs was higher in ATs from Polish patients as compared to French patients. This result can be explained by the overall higher environmental pollution in Poland, which means greater exposure on PAHs [41,42]. We can suppose that such high levels of PAHs in ATs are mainly caused by high air pollution in Krakow. According of WHO, Krakow is in the top ten percent of the most polluted cities in Europe. In addition, Polish patients had a higher BMI index meaning a higher content of body fat, which could lead to a greater accumulation of compounds. 

### 3.2. Naphthalene Is One of the Main PAHs Present in Human Blood Plasma and ATs and in French Patients Its Adipose Concentration Is Positively Correlated with BMI, Plasma Chemerin and Omental RARRES2 Gene Expression

The highest concentrations among detected PAHs were observed for naphthalene and phenanthrene, followed by anthracene, fluorene, and pyrene. These data are consistent with most of the literature data. Moon et al. noted the highest concentration of naphthalene and phenanthrene in the AT of Korean women [34]. Similarly, measurements of PAHs levels in pigs’ and cows’ tissues indicate again the highest accumulation in fat, but also that naphthalene and phenanthrene are the most abundant compounds [32]. On the other hand, in human blood serum samples collected during autopsy, predominant compounds were benzo(a)pyrene, benzo(b)fluoranthene and benzo(k)fluoranthene [9]. In the blood of children from the Indian city of Lucknow, naphthalene was the most abundant compound, while benzo(a)pyrene the least [8]. Thus, the abundance of the PAHs is dependent on the tissue. Differences in the accumulation of PAHs may result also from the chemical structure of these compounds. For example, in mice, it was observed that PAHs with a higher molecular weight and lower water solubility were less likely to bioaccumulate [31]. In French patients, we observed a significant positive correlation between the concentrations of naphthalene in all three types of AT with BMI. Our data are in good agreement with epidemiological studies that have shown an association between exposure to environmental chemicals including PAHs and an increased risk of obesity [20,45]. An association between urinary PAHs concentration and obesity in children has been also described [13,49]. Moreover, it has been shown in mice that exposure to benzo[a]pyrene, the most well-studied PAH, increases fat mass and leads to weight gain [50]. Some molecular mechanisms could explain these latter effects. For example, PAHs may alter serotonin signaling and impact feeding behaviors in rats [51] or reduce lipolysis directly through incorporation into adipocytes [52]. In contrast to French patients, we did not observe any correlation between AT naphthalene concentration and BMI in Polish patients. We think that the lack of significant correlations between BMI and naphthalene concentration in adipose tissue is due to the high variability of data concerning these two variables in Polish patients. This is especially true for BMI, whose values are significantly higher in this group of patients than in the French one. In addition to regulating metabolism and energy storage, AT secretes huge amounts of hormones, called adipokines, which affect the maintenance of proper homeostasis of the entire body. Thus, the level of these hormones reflects the proper functioning of AT [37]. We showed that the level of naphthalene in plasma and omental AT was positively correlated with the concentration of chemerin and resistin, and negatively with adiponectin. Thus, the observed correlations are mostly consistent with the physiological regulation of adipokine secretion by AT, since the level of adipokines correlates with the amount of AT, except for one: the expression of adiponectin decreases with increasing fat content [53,54]. Interestingly, we did not observe any significant relationship between the plasma concentrations of adipokines and PAHs in Polish patients that could be explained by the absence of correlations between BMI and PAH concentration in adipose tissues, and again the high variability of the data of these two variables in Polish patients.

### 3.3. Role of Naphthalene in Adipogenesis/Lipogenesis in 3T3-L1 Cells

A strong correlation between adipose naphthalene concentration with expression of adipokines in plasma and AT suggests that this PAH could be a factor favoring processes of adipogenesis/lipogenesis. Indeed, chemerin has been reported to increase during the differentiation of 3T3-L1 cells and human preadipocytes into adipocytes and play a key role in the adipogenesis and lipogenesis [55,56]. Adiponectin is an indicator of adipogenesis, and its production is dependent on the differentiation of adipocytes [57]. Resistin is a molecule that is secreted by adipocytes. These molecules have a critical role in causing obesity [58]. Adipogenesis, although a complex process controlled by many genes, briefly, begins with the proliferation of preadipocytes, followed by the accumulation of lipid droplets [44]. In this current study we indicated that naphthalene exposure increases the viability of preadipocytes of the 3T3-L1 line, but also stimulates their proliferation. Moreover, it has a significant impact on the differentiation of preadipocytes. Oil Red O staining of lipid drops clearly showed that a dose of 500 ng/mL of naphthalene increased the fat content by almost three-fold. The accumulation of lipid droplets is one of the main steps in the differentiation of pre-adipocytes into mature adipocytes [44]. In our study, the stimulatory effect of naphthalene on adipocyte differentiation was confirmed by the upregulated transcription of genes such as *Cebpa*, *Pparg*, *Adipoq*, which cooperate to orchestrate the completion of the full adipogenesis process [59,60]. Finally, the terminal stage of adipogenesis is represented by the induction of mature adipocyte genes, such as fatty acid synthase (FAS), acetyl-CoA carboxylase alpha (Acaca), fatty acid-binding protein 4 (FABP4) and perilipin, responsible for regulating adipocyte function and lipid droplet formation [61]. Naphthalene exposure increased Plin1 (perilipin), FABP4 and FAS gene expression. Due to the fact that chemerin and its receptor CMKLR1 are expressed in immature adipocytes, and they are highly involved in adipogenesis as well as glucose and lipid homeostasis [55], we checked the effect of naphthalene on chemerin secretion by 3T3-L1 cells in culture medium after differentiation. Naphthalene significantly increased the secretion of chemerin, which again confirms that naphthalene can promote adipogenesis in 3T3-L1 cells.

### 3.4. Limitations of Study 

The present study has some limitations.

Due to the small number of enrolled French and Polish patients this should be considered a pilot study.The Polish patients studied had a higher BMI than French. It is impossible to exclude the influence of this BMI difference on the parameters studied.The absence of adipokine expression data in the Polish patients (due to insufficient ATs collection) made it impossible to compare these data with French patients.Finally, it will be interesting to measure PAHs metabolites since they could be used as biomarkers of recent exposure to these compounds [10].

## 4. Materials and Methods

### 4.1. Study Group and Ethics Approval

A total of 20 subjects from the Clinical Hospital of Tours (*n* = 10) and from Department of General Surgery, Jagiellonian University Medical College (*n* = 10) were included in this pilot study. The Clinical Hospital of Tours is a reference center for the treatment of obesity in the Val de Loire Center region of France. The inclusion criteria were the following: patients with a BMI of more than 40 kg/m^2^ or with a BMI of 35–40 kg/m^2^. The non-inclusion criteria were patients with acute coronary syndrome, acute cerebrovascular accident in the last 2 months, decompensation of chronic heart failure, chronic kidney diseases, history of malignant neoplasms, presence of liver and thyroid diseases. The study was conducted according to principles set in the Declaration of Helsinki. Informed consent was signed by each participant and study protocol was approved by the Institutional Review Board. French patients were included in the prospective monocentric METABOSE cohort (Nutrition Department, CHU Tours) following patient written consent and after local ethical committee agreement (CNIL no. 18254562). Polish patients gave also written consent and after local ethical committee agreement (KBET number 1072.6120.94.2017).

### 4.2. Human Samples Collection

Human blood plasma and three types of white adipose tissue were collected from patients (*n* = 20) undergoing bariatric surgery. The different adipose tissue samples (1 mg) were obtained within less than 30 min after the incision from the following sites: subcutaneous abdominal close to the umbilicus; omental (epiploon) at the periphery of the omentum major; mesenteric at midgut close to the bowel.

### 4.3. PAHs Measurements in Blood Plasma and Adipose Tissues

A weighed sample of the tested material was freeze-dried at approximately −50°C. Prior to the extraction process, 2 µL of PAH-MIX 9 deuterated PAH standard at a concentration of 1 µg/mL was added to the sample. The extraction process was carried out in a Soxhlet extractor using a hexane/acetone mixture (1:1, *v*/*v*) for approximately 10 h. After this time, the volume of the extract solvent was reduced to a volume of about 5 mL using a rotary vacuum evaporator. The solution was subjected to initial purification by dialysis with semi-permeable membranes. Hexane was used as the dialysate. The process was carried out for 24 h. The excess dialysate solution was reduced to a volume of approximately 200 μL. In order to purify the extracts, tulip-shaped glass columns with solid packing of neutral silica gel (0.50 g) were used. Then, 200 µL of sample solution was applied to the top of the packing, previously washed with hexane. The analytes were eluted with a 40% solution of dichloromethane in hexane until 2 mL of the solution were collected. After evaporation of the excess solvent, the final volume of the extract was 50 µL in nonane. The determination of 16 polycyclic aromatic hydrocarbons was performed using a gas chromatograph coupled with a tandem mass spectrometer (TSQ 8000 Evo, Thermofisher, Poznan, Poland). Detailed reaction conditions are included in Table 8. The isotopically labeled internal standard method was used for quantification. 

The analysis of a PAH 50 ng/mL calibration standard containing a mixture of 16 PAHs, both natural and deuterated, was performed by introducing 1 µL of the standard into the chromatographic column. The chromatographic analysis of the samples was performed under the same conditions as the analysis of the standards by introducing 1 µL of the sample into the chromatographic column.

### 4.4. Adipokines Concentration in Human Blood Plasma 

Adiponectin, chemerin, visfatin and resistin concentrations were measured by commercially available ELISA assays in blood plasma samples. ELISA R&D Bio-Techne Ltd. kits (Abingdon, UK, Intra-assay coefficients of variations < 7% and inter-assay coefficients of variations ≤ 7%) were used for all adipokines.

### 4.5. The Culture and Treatment of 3T3-L1 Preadipocytes

The 3T3-L1 murine preadipocytes were purchased from American Type Culture Collection (Rockville, MD, United States), and were cultured in Dulbecco’s Modified Eagle’s Medium (DMEM, Eurobio, Les Ulis, France) with 10 % (*v*/*v*) fetal bovine serum and 100 U/mL penicillin, and 100 mg/mL streptomycin. To induce 3T3-L1 adipocyte differentiation, the post-confluent 3T3-L1 preadipocytes were cultured in DMEM with the insulin cocktail (10 μg/mL insulin, 1 μM dexamethasone, 0.5 mM isobutylmethylxanthine) for 2 days, and then the cells were cultured in DMEM with 10 μg/mL insulin for another 2 days. Beginning on day 4, the cells were cultured in normal DMEM for 4 days and the medium was freshly replaced every two days. To inhibit or facilitate 3T3-L1 adipocyte differentiation, 10–500 ng/mL naphthalene dissolved in ethanol (cat.no. 147141, Sigma, Saint Louis, MO, USA) was added throughout the differentiation period. Doses of naphthalene were chosen based on the concentration observed in ATs. Differentiation was verified by the appearance of fat droplets in the adipocytes [62].

### 4.6. Preadipocytes 3T3-L1 Cells Viability

3T3-L1 preadipocytes were plated in a 96-well plate with a density of 50 × 10^3^/well. After 24 h, the medium was changed to 10% FBS DMEM/F12 with naphthalene at different concentrations (1, 10, 100 and 500 ng/mL). Cell viability was determined after 24 h by using the Cell Count Kit-8 assay (cat. no. K1018, ApexBio Technology, Houston, TX 77054, USA). Briefly, 10 μL of 2-(methoxy-4-nitrophenyl)-3-(4-nitrophenyl)-5-(2,4-disulfophenyl)2H-tetrazolium (WST-8) solution was added into each well and cultured for 3 h in the incubator. Finally, the optical density (OD) value was obtained by using a spectrophotometer (TECAN, Männedorf, Switzerland) at a wavelength of 450 nm.

### 4.7. Preadipocytes 3T3-L1 Cells Proliferation

Cell proliferation was assessed using the BrdU assay (cat. no. 11647229001, Roche Diagnostics GmbH, Mannheim, Germany). A 96-well plate was used to carry out the experiment. A quadruple determination was carried out for each batch. The 30 × 10^3^ 3T3-L1 preadipocytes cells were seeded into each well. After 24 h, cells were treated with 1, 10, 100 and 500 ng/mL of naphthalene. After an incubation of a further 24 h, BrdU was added (20 μL/well). The further procedure was based on the information provided by provided by manufacturer’s instruction. The absorption was then measured at a wavelength of 450 nm on the spectral photometer (TECAN, Männedorf, Switzerland).

### 4.8. Oil Red O Staining

The Oil Red O powders (Sigma, Saint-Quentin-Fallavier, France) were dissolved in 1, 2-propanediol and prepared into 0.5% (*m*/*v*) Oil Red O solution. Before staining, Oil Red O solution was filtered through a 0.22 μm syringe filter. Preadipocytes were differentiated and treated in the absence or presence of 10 or 500 ng/mL of naphthalene in 6-well plates (10^6^ cells/well). The plates were washed with PBS and fixed with a 10% formalin solution (37% formaldehyde) in PBS and then incubated for 4 h in a diluted and filtered Oil Red O staining solution. The dye was removed, and the stained cells were washed twice with distilled water and photographed using the light microscope (Nikon, Champigny sur Marne, France). The Oil red O dye in fat droplets was dissolved in 60% isopropanol and lipid content was quantified at 510 nm in Tecan microplate reader (Bio-Tek Industries, Atlanta, GA, USA).

### 4.9. Real-Time Fluorescence Quantitative PCR Analysis

3T3L1 cells were seeded at density of 10^6^ cells/well in 6-well plates following the preadipocyte differentiation protocol. Samples were collected on day 8 after differentiation induction. The total RNA from differentiated 3T3-L1 cells incubated with or without different concentrations of naphthalene (10 or 500 ng/mL) was extracted by using the Nucleospin RNA Plus kit (Macherey-Nagel, Saint Quentin Fallavier, France) according to the manufacturer’s recommendations. Generation of cDNAs by reverse transcription, real-time PCR, and analysis of the relative expression of the gene of interest was performed, as previously described [63]. The primers of interest (*Fasn*, *Acaca*, *Cebpa*, *Pparg, Fabp4, Plin1, Rarres2, Nampt, Adipoq*) and reference genes (*Gapdh, Bactin and Ppia*) used in our study are listed in Table 9. The use of the geometric mean of several reference genes has already been validated [64].

### 4.10. Statistical Analyses

Data are shown as mean ± standard error of means (SEM) form at least three independent experiments. After examination on normal distribution, statistical analyses were carried out using the Student’s *t*-test or one way ANOVA. *p* ≤ 0.05 was considered as the statistically significant difference. To determine correlations between concentrations of the chosen PAHs in plasma or adipose tissue and BMI, adipokine concentration and expression were analyzed with the Pearson correlation coefficient, two-tailed *p* ≤ 0.05.

## 5. Conclusions

In conclusion, we detected the presence of all 16 priority PAHs in the plasma and adipose tissues of bariatric patients from France and Poland. Most of the PAHs had similar concentration in the three types of ATs (mesenteric, subcutaneous and omental). We observed significant correlation between naphthalene levels and patients BMI, as well as with chemerin, adiponectin and resistin plasma concentration and omental adipose tissue gene expression in French patients. Moreover, we demonstrated for the first time that naphthalene has a positive direct effect on adipocyte proliferation and differentiation and endocrine function by using in vitro studies on the murine 3T3-L1 cell line. Thus, naphthalene could be an obesogenic molecule and increase the risk of human obesity.

## Figures and Tables

**Figure 1 ijms-24-01455-f001:**
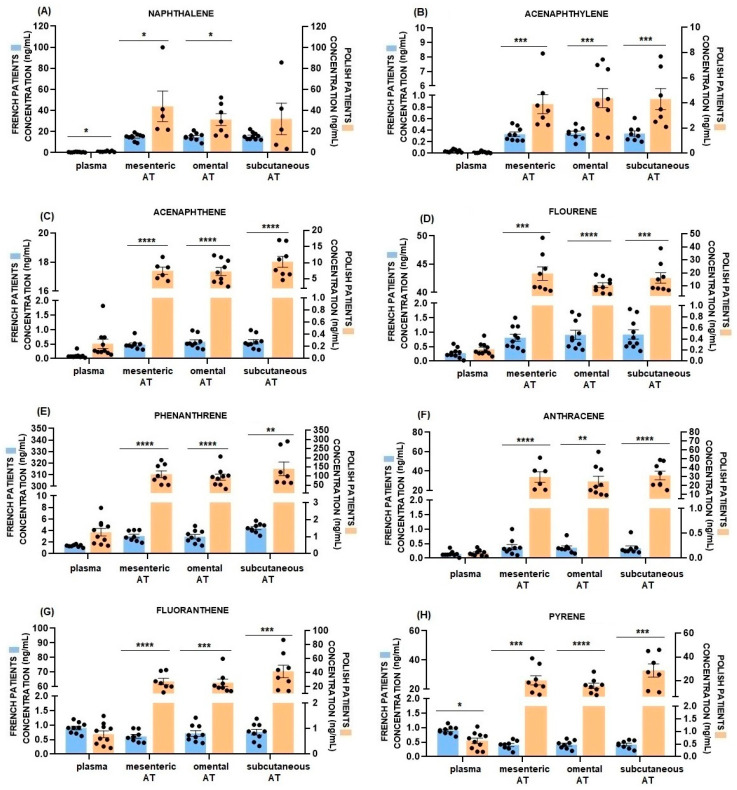
Comparison of PAH concentration in blood plasma and three types of AT between French (blue bar chart) and Polish patients (orange bar chart) (**A**–**P**) panel. Data are given as mean ± SEM, * *p* ≤ 0.05; ** *p* ≤ 0.01; *** *p* ≤ 0.001 and **** *p* ≤ 0.0001.

**Figure 2 ijms-24-01455-f002:**
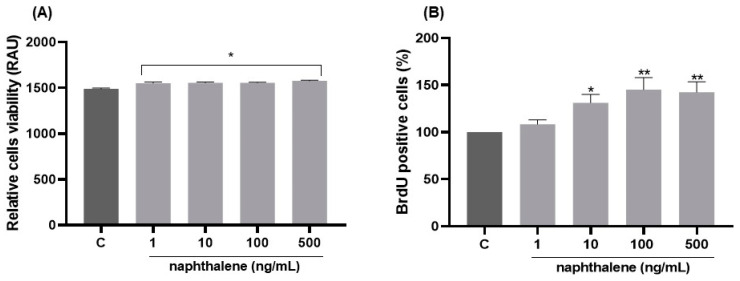
Dose-dependent effect of naphthalene on cell viability (**A**) and proliferation (**B**) of 3T3-L1 preadipocytes. 3T3-L1 preadipocytes were incubated without or with naphtalene in different concentrations for 24 h. For BrDU-positive cells, results were normalized to a percentage of control. Values are expressed as mean ± SEM (*n* = 7). * *p* < 0.05, ** *p* < 0.01 vs. control.

**Figure 3 ijms-24-01455-f003:**
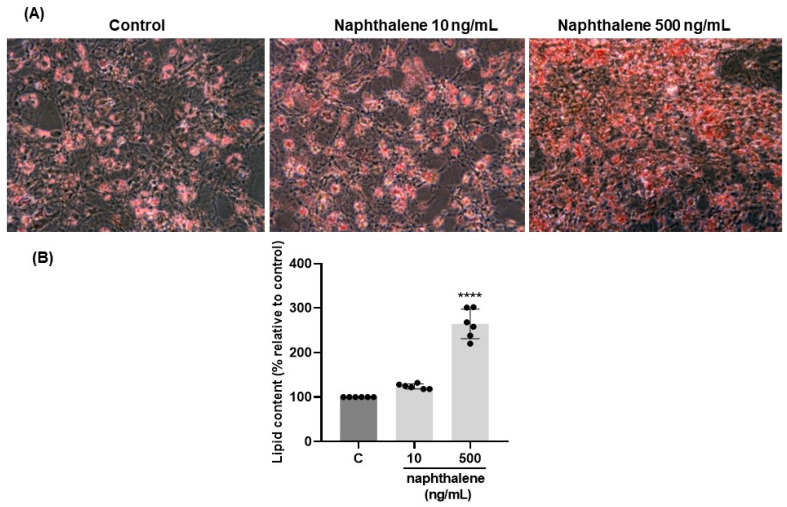
Effect of naphthalene on 3T3-L1 cell differentiation. (**A**): Representative picture of 3T3-L1 cells in response to naphtalene (10 and 500 ng/mL) after 7 days of differentiation stained with Oil Red O. (**B**): The values were normalized to the average of control group levels. Results are expressed as mean ± SEM (*n* = 6) from three independent experiments. **** *p* < 0.0001.

**Figure 4 ijms-24-01455-f004:**
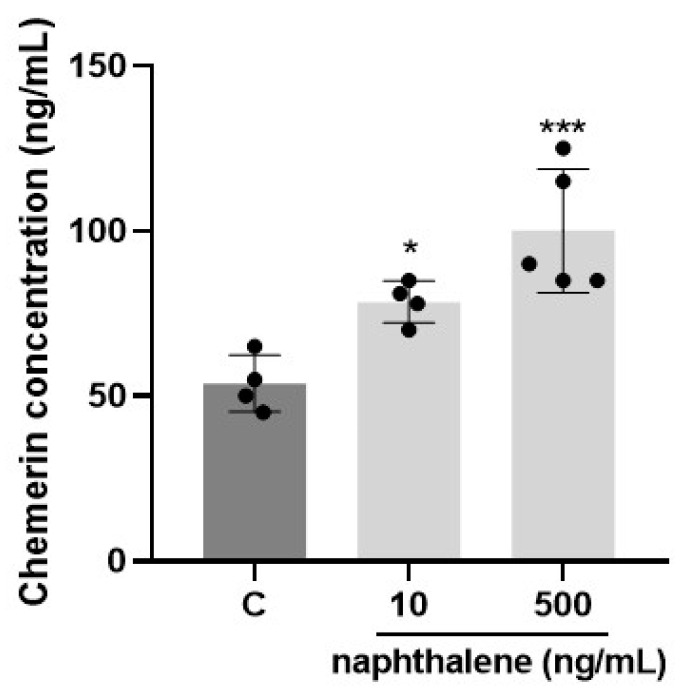
Effect of naphthalene on chemerin secretion by 3T3-L1 cells. Results are expressed as mean ± SEM (*n* = 6) from three independent experiments. The means of each group were compared to those of control group. * *p* < 0.05, *** *p* < 0.001.

**Table 1 ijms-24-01455-t001:** Characteristics of the study subjects. BMI, body mass index. Data are given as mean ± SEM. *p* ≤ 0.05 French patients (*n* = 10) vs. Polish patients (*n* = 10).

	French Patients (*n* = 10)	Polish Patients (*n* = 10)	*p* Value
Age (years)	45 ± 4.6	37 ± 2	*p* = 0.065
BMI (kg/m^2^)	40.2 ± 1.7	49 ± 2.5	*p* = 0.011
plasma glucose (mmol/L)	4.93 ± 0.20	5.37 ± 0.26	*p* = 0.193
plasma insulin (pmol/L)	70.9 ± 6.63	73.90 ± 4.42	*p* = 0.711
plasma chemerin (ng/mL)	220 ± 22.81	235.65 ± 34.66	*p* = 0.710
plasma adiponectin (µg/mL)	4.98 ± 0.38	4.15 ± 0.51	*p* = 0.208
plasma visfatin (ng/mL)	7.79 ± 1.05	7.55 ± 0.40	*p* = 0.833
plasma resistin (ng/mL)	10.11 ± 1.13	5.78 ± 0.47	*p* = 0.002

**Table 2 ijms-24-01455-t002:** Comparison of PAH concentrations between plasma (ng/mL) and three white ATs (mesenteric, omental, and subcutaneous; ng/g) in French patients (*n* = 10). Data are given as mean ± SEM, *p* ≤ 0.05. Different capital letters indicate a significant effect between different ATs and blood plasma whereas lower case letters indicate a significant effect between the 16 PAHs.

	Plasma	Mesenteric AT	Omental AT	Subcutaneous AT	*p* Value
naphthalene	0.52 ± 0.09 ^b,A^	14.76 ± 0.96 ^c,B^	15.10 ± 1.14 ^c,B^	15.60 ± 1.05 ^c,B^	*p* < 0.0001
acenaphthylene	0.04 ± 0.01 ^a,A^	0.33 ± 0.04 ^a,B^	0.34 ± 0.03 ^a,B^	0.34 ± 0.04 ^a,B^	*p* < 0.0001
acenaphthene	0.10 ± 0.03 ^a,A^	0.49 ± 0.05 ^a,B^	0.57 ± 0.07 ^a,B^	0.58 ± 0.07 ^a,B^	*p* < 0.0001
fluorene	0.28 ± 0.05 ^a,A^	0.81 ± 0.12 ^a,B^	0.91 ± 0.15 ^a,B^	0.92 ± 0.16 ^a,B^	*p* = 0.0032
phenanthrene	1.36 ± 0.07 ^d,A^	3.05 ± 0.28 ^b,B^	2.95 ± 0.36 ^b,B^	4.43 ± 0.25 ^b,C^	*p* < 0.0001
anthracene	0.15 ± 0.03 ^a,A^	0.38 ± 0.09 ^a,A^	0.35 ± 0.06 ^a,A^	0.35 ± 0.07 ^a,A^	*p* = 0.0689
fluoranthene	0.90 ± 0.06 ^c,A^	0.61 ± 0.06 ^a,A^	0.71 ± 0.09 ^a,A^	0.76 ± 0.09 ^a,A^	*p* = 0.1366
pyrene	0.91 ± 0.04 ^c,B^	0.40 ± 0.05 ^a,A^	0.42 ± 0.05 ^a,A^	0.42 ± 0.04 ^a,A^	*p* < 0.0001
benz(a)anthracene	0.05 ± 0.02 ^a,A^	0.14 ± 0.02 ^a,B^	0.17 ± 0.02 ^a,B^	0.16 ± 0.02 ^a,B^	*p* < 0.0001
chrysene	0.06 ± 0.02 ^a,A^	0.14 ± 0.02 ^a,B^	0.16 ± 0.02 ^a,B^	0.17 ± 0.01 ^a,B^	*p* = 0.0007
benzo(b)fluoranthene	0.11 ± 0.03 ^a,A^	0.21 ± 0.02 ^a,B^	0.24 ± 0.02 ^a,B^	0.24 ± 0.02 ^a,B^	*p* = 0.0012
benzo(k)fluoranthene	0.06 ± 0.02 ^a,A^	0.15 ± 0.02 ^a,B^	0.18 ± 0.02 ^a,B^	0.17 ± 0.01 ^a,B^	*p* < 0.0001
benzo(a)pyrene	0.21 ± 0.06 ^a,A^	0.16 ± 0.02 ^a,A^	0.19 ± 0.02 ^a,A^	0.19 ± 0.02 ^a,A^	*p* = 0.8091
indeno(1,2,3-cd)pyrene	0.06 ± 0.02 ^a,A^	0.16 ± 0.01 ^a,B^	0.19 ± 0.01 ^a,B^	0.17 ± 0.01 ^a,B^	*p* < 0.0001
dibenz(a,h)anthracene	0.04 ± 0.01 ^a,A^	0.11 ± 0.01 ^a,B^	0.14 ± 0.01 ^a,B^	0.13 ± 0.01 ^a,B^	*p* < 0.0001
benzo(g,h,i)perylene	0.04 ± 0.01 ^a,A^	0.15 ± 0.01 ^a,B^	0.17 ±0.01 ^a,B^	0.16 ± 0.01 ^a,B^	*p* < 0.0001
	*p* < 0.0001	*p* < 0.0001	*p* < 0.0001	*p* < 0.0001	

**Table 3 ijms-24-01455-t003:** Comparison of PAH concentrations between plasma (ng/mL) and three white adipose tissues (mesenteric, omental, and subcutaneous; ng/g) in Polish patients (*n* = 10). Data are given as mean ± SEM, *p* ≤ 0.05. Different capital letters indicate a significant effect between different ATs and blood plasma whereas lower case letters indicate a significant effect between the 16 PAHs.

	Plasma	Mesenteric AT	Omental AT	Subcutaneous AT	*p* Value
naphthalene	1.07 ± 0.18 ^c,A^	44.19 ± 10.28 ^d,B^	31.56 ± 4.66 ^d,B^	50.02 ± 16.75 ^e,B^	*p* = 0.0101
acenaphthylene	0.07 ± 0.02 ^a,A^	3.90 ± 0.31 ^ab,B^	4.38 ± 0.72 ^b,B^	4.31 ± 0.70 ^b,B^	*p* < 0.0001
acenaphthene	0.24 ± 0.08 ^a,A^	7.47 ± 0.63 ^b,B^	7.29 ± 1.16 ^bc,B^	10.25 ± 1.51 ^bc,B^	*p* < 0.0001
fluorene	0.22 ± 0.04 ^a,A^	19.49 ± 5.05 ^c,B^	10.68 ± 1.74 ^c,B^	16.13 ± 3.64 ^c,B^	*p* = 0.0007
phenanthrene	1.26 ± 0.22 ^c,A^	109.94 ± 14.37 ^e,B^	93.99 ± 16.31 ^e,B^	139.43 ± 31.47 ^f,B^	*p* < 0.0001
anthracene	0.11 ± 0.02 ^a,A^	29.69 ± 4.89 ^c,B^	24.97 ± 5.39 ^d,B^	42.00 ± 4.38 ^e,B^	*p* < 0.0001
fluoranthene	0.78 ± 0.13 ^b,A^	27.52 ± 3.40 ^c,B^	25.63 ± 4.80 ^d,B^	49.44 ± 10.31 ^e,B^	*p* < 0.0001
pyrene	0.63 ± 0.11 ^b,A^	20.08 ± 2.74 ^c,BC^	19.39 ± 4.20 ^d,B^	28.43 ± 4.63 ^d,C^	*p* < 0.0001
benz(a)anthracene	0.03 ± 0.01 ^a,A^	0.97 ± 0.18 ^a,B^	1.11 ± 0.29 ^ab,B^	0.89 ± 0.19 ^a,B^	*p* = 0.0022
chrysene	0.05 ± 0.01 ^a,A^	0.38 ± 0.09 ^a,B^	0.35 ± 0.06 ^a,B^	0.37 ± 0.06 ^a,B^	*p* = 0.0023
benzo(b)fluoranthene	0.08 ± 0.02 ^a,A^	0.44 ± 0.08 ^a,B^	0.54 ± 0.12 ^a,B^	0.39 ± 0.09 ^a,AB^	*p* = 0.0044
benzo(k)fluoranthene	0.05 ± 0.01 ^a,A^	0.43 ± 0.08 ^a,B^	0.41 ± 0.09 ^a,B^	0.31 ± 0.05 ^a,AB^	*p* = 0.0022
benzo(a)pyrene	0.03 ± 0.003 ^a,A^	0.17 ± 0.01 ^a,AB^	0.25 ± 0.05 ^a,B^	0.34 ± 0.07 ^a,B^	*p* = 0.0004
indeno(1,2,3-cd)pyrene	0.05 ± 0.01 ^a,A^	0.33 ± 0.11 ^a,A^	0.30 ± 0.08 ^a,A^	0.25 ± 0.07 ^a,A^	*p* = 0.0662
dibenz(a,h)anthracene	0.08 ± 0.01 ^a,A^	0.37 ± 0.09 ^a,AB^	0.49 ± 0.09 ^a,B^	0.40 ± 0.09 ^a,AB^	*p* = 0.0091
benzo(g,h,i)perylene	0.06 ± 0.01 ^a,A^	0.35 ± 0.08 ^a,AB^	0.56 ± 0.08 ^a,B^	0.52 ± 0.13 ^a,B^	*p* = 0.0019
	*p* < 0.0001	*p* < 0.0001	*p* < 0.0001	*p* < 0.0001	

**Table 4 ijms-24-01455-t004:** Correlation between BMI and PAH concentration in ATs (ng/g) or blood plasma (ng/mL) in (A) French (*n* = 10) and Polish (*n* = 10) patients.

	Patients	Naphthalene	Phenanthrene	Fluoranthene	Pyrene
Plasma	French	r = −0.303 *p* = 0.395	r = 0.293 *p* = 0.411	r = 0.273 *p* = 0.445	r = 0.254 *p* = 0.479
	Polish	r = 0.613 *p* = 0.106	r = 0.504 *p* = 0.202	r = 0.442 *p* = 0.273	r = 0.389 *p* = 0.341
Mesenteric AT	French	r = 0.743 *p* = 0.014	r = −0.141 *p* = 0.698	r = −0.418 *p* = 0.229	r = −0.366 *p* = 0.298
	Polish	r = 0.299 *p* = 0.515	r = −0.664 *p* = 0.104	r = −0.701 *p* = 0.079	r = −0.675 *p* = 0.096
Omental AT	French	r = 0.904 *p* = 0.0003	r = −0.198 *p* = 0.583	r = −0.368 *p* = 0.295	r = −0.425 *p* = 0.221
	Polish	r = 0.508 *p* = 0.199	r = −0.426 *p* = 0.292	r = −0.379 *p* = 0.354	r = −0.362 *p* = 0.378
Subcutaneous AT	French	r = 0.906 *p* = 0.0003	r = −0.228 *p* = 0.527	r = −0.285 *p* = 0.425	r = −0.393 *p* = 0.262
	Polish	r = 0.355 *p* = 0.489	r = −0.241 *p* = 0.602	r = −0.383 *p* = 0.396	r = −0.314 *p* = 0.493

**Table 5 ijms-24-01455-t005:** Correlation between PAH concentration and adipokines level in blood plasma of French (*n* = 10) and Polish (*n* = 10) patients.

	Patients	Naphthalene	Phenanthrene	Fluoranthene	Pyrene
Chemerin	French	r = 0.979 *p* < 0.0001	r = −0.313 *p* = 0.379	r = −0.435 *p* = 0.209	r = −0.309 *p* = 0.385
	Polish	r = 0.086 *p* = 0.812	r = 0.510 *p* = 0.132	r = 0.427 *p* = 0.218	r = 0.352 *p* = 0.319
Adiponectin	French	r = −0.938 *p* < 0.0001	r = 0.277 *p* = 0.439	r = 0.424 *p* = 0.222	r = 0.461 *p* = 0.180
	Polish	r = −0.078 *p* = 0.831	r = −0.490 *p* = 0.150	r = −0.436 *p* = 0.208	r = −0.424 *p* = 0.222
Visfatin	French	r = 0.174 *p* = 0.630	r = 0.171 *p* = 0.636	r = 0.042 *p* = 0.908	r = −0.118 *p* = 0.745
	Polish	r = 0.186 *p* = 0.608	r = −0.363 *p* = 0.302	r = −0.406 *p* = 0.244	r = −0.422 *p* = 0.224
Resistin	French	r = 0.838 *p* = 0.002	r = −0.158 *p* = 0.663	r = −0.317 *p* = 0.372	r = −0.408 *p* = 0.241
	Polish	r = 0.098 *p* = 0.787	r = 0.327 *p* = 0.356	r = 0.295 *p* = 0.408	r = 0.285 *p* = 0.424

**Table 6 ijms-24-01455-t006:** Correlation between PAH concentration and adipokines mRNA level in omental AT of French patients (*n* = 10).

	Naphthalene	Phenanthrene	Fluoranthene	Pyrene
RARRES2	r = 0.679 *p* = 0.031	r = 0.070 *p* = 0.848	r = −0.179 *p* = 0.645	r = −0.419 *p* = 0.229
ADIPOQ	r = −0.635 *p* = 0.049	r = 0.107 *p* = 0.768	r = −0.165 *p* = 0.671	r = −0.379 *p* = 0.280
NAMPT	r = 0.323 *p* = 0.362	r = −0.199 *p* = 0.582	r = −0.439 *p* = 0.237	r = −0.427 *p* = 0.218
RETN	r = 0.667 *p* = 0.035	r = 0.047 *p* = 0.897	r = −0.264 *p* = 0.492	r = −0.427 *p* = 0.218

**Table 7 ijms-24-01455-t007:** Effect of naphthalene on gene expression of factors involved in adipogenesis and lipogenesis in 3T3-L1 cells. Values are expressed as mean ± SEM (*n* = 3), *p* ≤ 0.05; different capital letters indicate a significant difference between control and different concentrations of naphthalene.

			Naphthalene (ng/mL)	
	Gene	C	10	500
Lipogenic enzymes	*Fasn*	1.22 ± 0.06 ^A^	21.08 ± 2.86 ^B^	135.46 ± 10.35 ^C^
	*Acaca*	1.48 ± 0.09 ^A^	8.22 ± 0.47 ^B^	34.32 ± 2.30 ^C^
Pro-adipogenesis factor	*Cebpa*	2.12 ± 0.20 ^A^	6.16 ± 0.27 ^B^	51.06 ± 3.64 ^C^
	*Pparg*	1.38 ± 0.11 ^A^	32.66 ± 3.64 ^B^	113.16 ± 4.72 ^C^
	*Fabp4*	1.68 ± 0.14 ^A^	5.38 ± 0.28 ^B^	20.46 ± 1.62 ^C^
	*Plin1*	1.78 ± 0.15 ^A^	4.02 ± 0.50 ^B^	15.36 ± 0.72 ^C^
Adipokines	*Rarres2*	1.22 ± 0.06 ^A^	2.52 ± 0.22 ^B^	11.88 ± 0.51 ^C^
	*Adipoq*	1.22 ± 0.06 ^A^	4.54 ± 0.47 ^B^	10.42 ± 0.62 ^C^
	*Retn*	1.22 ± 0.06 ^A^	4.18 ± 0.59 ^B^	9.84 ± 0.92 ^C^

**Table 8 ijms-24-01455-t008:** Operating conditions of the apparatus.

Carrier Gas Type	Helium
Dispenser operating temperature	260 °C
Detector operating temperature	260 °C
Volume of dispensed calibration standard PAH 50 ng	1 µL
Dispensed sample volume	1 µL
Transfer line	280 °C
Ion source	EI; 280 °C
Chromatography column	DB5 MS 30 m, diameter 0.250 mm, phase thickness 0.25 µm
Temperature programme	60 °C for 3 min, 15 °C/min ramp rate up to 280 °C, 280 °C for 25 min

**Table 9 ijms-24-01455-t009:** Primers used for the RTqPCR.

Gene Symbol	Primers Sequences	Product Length	GenBank ID
GAPDH	5′-TGCACCACCAACTGCTTAGC-3′ 5′-GGATGCAGGGATGATGTTCT-3′	177	NM_008084.2
BACTIN	5′-CCTGTGCTGCTCACCGAGGC-3′ 5′-GACCCCGTCTCTCCGGATCCATC-3′	174	NM_007393
PPIA	5′-CGCTTGCTGCAGCCATGGTC-3′ 5′-CAGCTCGAAGGAGACGCGGC-3′	86	NM_008907.1
FASN	5′-GGAGGTGGTGATAGCCGGTAT-3′ 5′-TGGGTAATCCATAGAGCCCAG-3′	139	XM030245556.1
ACACA	5′-CGATCTATCCGTCGGTGGTC-3′ 5′-TATTCTGCATTGGCTTTAAG-3′	100	XM011248667.2
CEBPA	5′-CAAGAACAGCAACGAGTACCG-3′ 5′-GTCACTGGTCAACTCCAGCAC-3	128	NM001287514.1
PPARG	5′-GGAAGACCACTCGCATTCCTT-3′ 5′-GTAATCAGCAACCATTGGGTCA-3′	120	XR001785108.2
FABP4	5′-AAGGTGAAGAGCATCATAACCCT-3′ 5′-TCACGCCTTTCATAACACATTCC-3′	129	NM001409513.1
PLIN1	5′-CACTCTCTGGCCATGTGGA-3′ 5′-AGCCAGGGCACCCGCACCTC-3′	118	NM175640.2
RARRES2	5′-GACCAACTGCCCCAAGAA-3′ 5′-GTCCATTTTAATGCAGGCCAG-3′	93	NM001347168.1
NAMPT	5′-GAATGTCTCCTTCGGTTCTGG-3′ 5′-TCAGCAACTGGGTCCTTAAAC-3′	111	NM021524.2
ADIPOQ	5′-TGTCTGTACGATTGTCAGTGG-3′ 5′-GCAGGATTAAGAGGAACAGGAG-3′	86	NM 009605.5
RETN	5′-AGACTGCTGTGCCTTCTGGG-3′ 5′-CCCTCCTTTTCCTTTTCTTCCTTG-3′	200	NM022984.4

## Data Availability

All the data are provided in the manuscript.

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
