# Peer review of "Concentration of Polycyclic Aromatic Hydrocarbons (PAHs) in Human Serum and Adipose Tissues and Stimulatory Effect of Naphthalene in Adipogenesis in 3T3-L1 Cells"

_ijms, 2023, doi:10.3390/ijms24021455_

Round 1

Reviewer 1 Report

This study confirms distribution of polycyclic aromatic hydrocarbons (PAHs) in adipose tissues and clarify the effects of naphthalene on proliferation and differentiation of preadipocytes. The results are suggestive in considering the relationship between PAHs and AT. I hope the research will be developed further. 

Some comments are as follows.

Explanations and discussions

Table 2 & Table 3 and Figure 1

Same results are shown in Table and Figure. For paper, only Figure 1 is enough. On the other hand, the actual values are informative. So Table 2 and Table 3 can be attached as supporting information.

Table 3

Line 136-139. The explanation “the concentrations did not differ significantly between plasma and mesenteric or subcutaneous AT” seems complicated, between plasma concentrations, between plasma and mesenteric concentrations, between plasma and subcutaneous concentrations, and so on. Concentrations in plasma look to be apparently lower than those in ATs for the PAHs mentioned in this sentence.

Figure 1.

Figure A-H is hidden under Figure I-P, so I cannot confirm Figure A-H.

-       Only the left-side vertical axis is enough to show the results. The right-side axis is not necessary.

-       The difference in color can be explained in the figure legend.

-       Each title is very helpful, although “CONCNETRATION OF” and “(ng/mL)” should be shown beside the vertical axis, not with the title.

Table 4

Line 322. The meaning of “This could be explained by the higher BMI of Polish patient’s as compared to French patients” does not seem to be clear. More detailed explanation is necessary to why correlation between AT naphthalene concentration and BMI cannot be observed in persons with higher BMI.

Line 331-333. More detailed explanation is necessary to why no significant relationship between the plasma concentrations of adipokines and PAHs in Polish patients could be explained by the higher BMI and PAH levels in Polish patients.

Figure 2.

Relative cell viabilities look to be same for all cells treated with different concentration of naphthalene, while higher concentration of naphthalene apparently stimulated cell proliferation. Why the relative cell viabilities were same for all the concentration of naphthalene?

Miss typing

Line 41. “AT on” should be “AT, on”.

Line 62. “benzo(a)anthracene” is already mentioned and should be deleted.

Line 62. “benzo(a)fluoranthene” should be “benzo(b)fluoranthene”.

Line 63. “benzo(g),h,i)perylene” should be “benzo(g,h,i)perylene”.

Lien 85. “[29] environmental” should be “[29], environmental”.

Line 86. “asother” should be “as other”.

Line 88. “AT” should be “adipose tissue (AT)”.

Line 105. “(Malopol” should be “Malopol”.

Line 107. “what” should be “which”.

Line 138, Table 2 and Table 3. “benzo(ghi)perylene” should be “benzo(g,h,i)perylene”.

Line 156. “patients” should be “patients,”.

Line 161. “to” should be deleted.

Line 197. “patients’” should be “French patients’”.

Line 200. “24h” should be “24 h”.

Line 215. “red oil” should be “Oil Red O powders”.

Line 222. “rarres2” should be “Rarres2”.

Line 252. “RARRES2” should be “Rarres2”.

Line 254. “cells” should be deleted.

Line 280. “their involved more in processes related to” should be deleted.

Line 285. “, only” should be “, although only”.

Line 293. “what it means” should be “meaning”.

Line 296. “one the main” should be “one of the main”.

Line 310. “higher weight” should be 2higher molecular weight”.

Line 319. “later” should be “latter”.

Line 322. “BMI” should be “BMI in Polish patients”.

Line 323. “patient’s” should be “patients”.

Line 330. “only” should be “except that only”.

Line 337. “plasma” should be “in plasma”.

Line 337. “favoured” should be “favouring”.

Line 341. “its” should be “and its”.

Line 342. “linking” be “in causing”.

Line 343. “and” should be deleted.

Line 344. “cells” should be deleted.

Line 347. “Red oil staining” should be “Oil Red O staining”.

Line 348. “500 ng/mL” should be “500 ng/mL of naphthalene”.

Line 352. “levels transcripts genes” should be “transcription of genes”.

Line 352. “Cebpa, Pparg, AdipoQ, PPARg and C/EBPa” should be “CebpaPparg and Adipoq”.

Line 352. “when expressed,” should be “which”.

Line 362. “its secretion” should be “the secretion of chemerin”.

Lin3 362. “it” should be “naphthalene”.

Line 405. “RVE (rotary vacuum evaporator)” should be “rotary vacuum evaporator”.

Line 406. The meaning of “SPM” is unclear. This word appears only once here, so it should be written in full spelling, not in abbreviation.

Line 411. Is “2 mL” correct? It seems too small.

Line 414. “GC/M” is not necessary.

Line 427. “Kingdom)” should be “Kingdom,”.

Line 459. “manufacturers” should be “manufacturer’s”

Line 468. “red O oil” should be “Oil Red O”.

Line 482. “references” should be “reference”.

Author Response

Review Report 1

This study confirms distribution of polycyclic aromatic hydrocarbons (PAHs) in adipose tissues and clarify the effects of naphthalene on proliferation and differentiation of preadipocytes. The results are suggestive in considering the relationship between PAHs and AT. I hope the research will be developed further. 

Some comments are as follows.

Explanations and discussions

Table 2 & Table 3 and Figure 1

Same results are shown in Table and Figure. For paper, only Figure 1 is enough. On the other hand, the actual values are informative. So Table 2 and Table 3 can be attached as supporting information.

Authors’ comment: We are agree with review 1 but we think that it is important to show the real values. So if you don’t mind please we prefer to let the tables with Figure 1.

Table 3

Line 136-139. The explanation “the concentrations did not differ significantly between plasma and mesenteric or subcutaneous AT” seems complicated, between plasma concentrations, between plasma and mesenteric concentrations, between plasma and subcutaneous concentrations, and so on. Concentrations in plasma look to be apparently lower than those in ATs for the PAHs mentioned in this sentence.

Authors’ comment: The concentration of the compounds mentioned in the sentence in Polish patients is lower in plasma, however, in some cases, the differences are not statistically significant. We have improved this sentence to make it clearer and to indicate exactly between which tissues and for which compounds no significant differences were observed.

Figure 1.

Figure A-H is hidden under Figure I-P, so I cannot confirm Figure A-H.

Authors’ comment: We are very sorry for this error. We have made the change so that both parts of figure 1 are visible

-       Only the left-side vertical axis is enough to show the results. The right-side axis is not necessary.

Authors’ comment: We added an axis on the left and right sides because for some PAHs, concentrations differ a lot between French and Polish patients, and must cut the axis in different places. We were unable to establish a single scale for the concentration of some PAHs for Polish and French patients together.

-       The difference in color can be explained in the figure legend.

Authors’ comment: In the description of Figure 1 we have included information about the meaning of the colours of the bars.

-       Each title is very helpful, although “CONCNETRATION OF” and “(ng/mL)” should be shown beside the vertical axis, not with the title.

Authors’ comment: As suggested by the reviewer we added “concentration of “ and (ng/ml)” beside the vertical axis and we let the name of PAH in the title.

Table 4

Line 322. The meaning of “This could be explained by the higher BMI of Polish patient’s as compared to French patients” does not seem to be clear. More detailed explanation is necessary to why correlation between AT naphthalene concentration and BMI cannot be observed in persons with higher BMI.

Authors’ comment: Thank you for your observation. We think that the lack of significant correlation between BMI and naphthalene concentration in adipose tissue is due to the high variability of data concerning these two variables in Polish patients. This is especially true for BMI, whose values are significantly higher in this group of patients than in French one.

Line 331-333. More detailed explanation is necessary to why no significant relationship between the plasma concentrations of adipokines and PAHs in Polish patients could be explained by the higher BMI and PAH levels in Polish patients.

Authors’ comment: Thank you for your comments. We have added more explanation to this statement. This question is related to the previous one.

Figure 2.

Relative cell viabilities look to be same for all cells treated with different concentration of naphthalene, while higher concentration of naphthalene apparently stimulated cell proliferation. Why the relative cell viabilities were same for all the concentration of naphthalene?

Authors’ comment: Thank you for your comments. It is an interesting question. In our research, we used the BrdU test to assess the proliferation of 3T3-L1 cells, which clearly showed that naphthalene stimulates the proliferation of preadipocytes dose-dependent. To assess the viability of these cells, we used a CCK8 test based on the metabolic activity of the cells, and we did not observe a dose-dependent effect here. This could explain by the different sensitivity of the two tests used (CCK8 and BrDU).

Miss typing

Line 41. “AT on” should be “AT, on”.

Line 62. “benzo(a)anthracene” is already mentioned and should be deleted.

Line 62. “benzo(a)fluoranthene” should be “benzo(b)fluoranthene”.

Line 63. “benzo(g),h,i)perylene” should be “benzo(g,h,i)perylene”.

Lien 85. “[29] environmental” should be “[29], environmental”.

Line 86. “asother” should be “as other”.

Line 88. “AT” should be “adipose tissue (AT)”.

Line 105. “(Malopol” should be “Malopol”.

Line 107. “what” should be “which”.

Line 138, Table 2 and Table 3. “benzo(ghi)perylene” should be “benzo(g,h,i)perylene”.

Line 156. “patients” should be “patients,”.

Line 161. “to” should be deleted.

Line 197. “patients’” should be “French patients’”.

Line 200. “24h” should be “24 h”.

Line 215. “red oil” should be “Oil Red O powders”.

Line 222. “rarres2” should be “Rarres2”.

Line 252. “RARRES2” should be “Rarres2”. Authors’ comment: In this sentence, we refer to the expression of the human gene for chemerin in human omental adipose tissue, therefore we have used the abbreviation RARRES2, according to the nomenclature.

Line 254. “cells” should be deleted.

Line 280. “their involved more in processes related to” should be deleted.

Line 285. “, only” should be “, although only”.

Line 293. “what it means” should be “meaning”.

Line 296. “one the main” should be “one of the main”.

Line 310. “higher weight” should be 2higher molecular weight”.

Line 319. “later” should be “latter”.

Line 322. “BMI” should be “BMI in Polish patients”.

Line 323. “patient’s” should be “patients”.

Line 330. “only” should be “except that only”.

Line 337. “plasma” should be “in plasma”.

Line 337. “favoured” should be “favouring”.

Line 341. “its” should be “and its”.

Line 342. “linking” be “in causing”.

Line 343. “and” should be deleted.

Line 344. “cells” should be deleted.

Line 347. “Red oil staining” should be “Oil Red O staining”.

Line 348. “500 ng/mL” should be “500 ng/mL of naphthalene”.

Line 352. “levels transcripts genes” should be “transcription of genes”.

Line 352. “Cebpa, Pparg, AdipoQ, PPARg and C/EBPa” should be “CebpaPparg and Adipoq”.

Line 352. “when expressed,” should be “which”.

Line 362. “its secretion” should be “the secretion of chemerin”.

Line 362. “it” should be “naphthalene”.

Line 405. “RVE (rotary vacuum evaporator)” should be “rotary vacuum evaporator”.

Line 406. The meaning of “SPM” is unclear. This word appears only once here, so it should be written in full spelling, not in abbreviation.

Line 411. Is “2 mL” correct? It seems too small.

Line 414. “GC/M” is not necessary.

Line 427. “Kingdom)” should be “Kingdom,”.

Line 459. “manufacturers” should be “manufacturer’s”

Line 468. “red O oil” should be “Oil Red O”.

Line 482. “references” should be “reference”.

Authors’ comment: Thank you for reading the manuscript carefully and for all your comments. We have corrected all of the indicated miss typings.

Reviewer 2 Report

The abstract is not well written. Please include a short background, how the study was done, brief results, and a conclusion. Please remove the detailed aim from the abstract. This can be added in the introduction, and it is already mentioned there.

There are some problems in Figure-1. Please make corrections.

Figure-3A, please include enlarged or magnified images to show them clearly.

The author also provided the limitations of their study, and it made the study easy to understand and will lead to future researchers to overcome the limitations.

If possible, please include a graphical abstract.

Author Response

Review Report 2

The abstract is not well written. Please include a short background, how the study was done, brief results, and a conclusion. Please remove the detailed aim from the abstract. This can be added in the introduction, and it is already mentioned there.

Authors’ comment: Thank you for your comment. The word limit given by the journal made it difficult for us to get a proper background in the abstract, but we made some corrections according to your suggestions and shortened the part about the research objectives.

Abstract: Polycyclic aromatic hydrocarbons (PAHs) are one of the most prevalent clas-ses of environmental pollutants. Some evidence shows that PAHs could be involved in human obesity. However, little is known about the distribution patterns of PAHs in human adipose tissue (AT) and the role of PAHs on adipogenesis/lipogenesis. The aims of this pilot study were to determine concentrations of 16 PAHs defined as High Priori-ty Pollutants in plasma and adipose tissue of French and Polish bariatric patients, and their correlation with body mass index (BMI) as well plasma and AT adipokines expression levels. We finally investigate the role of naphthalene on cell proliferation, vi-ability, and differentiation in 3T3-L1 preadipocytes. The concentration of most PAHs was similar in the three types of AT and it was significantly higher in AT as compared to plasma suggesting bioaccumulation. Polish patients had higher PAHs levels in AT than French ones. Only the concentration of naphthalene in AT was positively corre-lated with the BMI and serum or adipose chemerin, adiponectin and resistin expression in French but not in Polish patients that had significantly higher BMI. Moreover, naphthalene exposure increased cell proliferation of 3T3-L1 preadipocytes and lipo-genesis and expression genes involved in adipogenesis after cell differentiation. Taken together, PAHs and more particularly naphthalene could be an obesogenic molecule and increase the risk of obesity.

There are some problems in Figure-1. Please make corrections.

Authors’ comment: We are very sorry for this error. We have made the change and now both parts of Figure 1 are visible.

Figure-3A, please include enlarged or magnified images to show them clearly.

Authors’ comment: As suggested by the reviewer we included enlarged images to show them clearly.

The author also provided the limitations of their study, and it made the study easy to understand and will lead to future researchers to overcome the limitations.

Authors’ comment: We thank the reviewer for her/his comment.

If possible, please include a graphical abstract.

Authors’ comment: As suggested by the reviewer we added a graphical abstract.